# Equiflow: An open-source software package for evaluating changes in cohort composition

**Jacob Gould Ellen[1], Chrystinne Fernandes[2]\*, Martin Viola[1], Keagan Yap[3], Arinda Jordan[4], Mutesi Flavia Kirabo[4], João Matos[5], Pedro Moreira[2], Leo Anthony Celi[2]**

**1** Harvard Medical School, Boston, Massachusetts, United States of America, **2** Institute for Medical Engineering and Science, Massachusetts Institute of Technology, Cambridge, Massachusetts, United States of America, **3** Harvard College, Cambridge, Massachusetts, United States of America, **4** Mbarara University of Science and Technology, Mbarara, Uganda, **5** Centre for Statistics in Medicine, University of Oxford, Oxford, United Kingdom

\* cof@mit.edu

## Abstract

Clinical research studies routinely apply exclusion criteria and data preprocessing steps that can substantially alter dataset composition, potentially introducing hidden biases that affect validity and generalizability. This is particularly important in artificial intelligence/machine learning (AI/ML) studies where models learn patterns directly from training data. We developed Equiflow, an open-source Python package that automates creation of enhanced participant flow diagrams tracking both sample size and composition changes throughout studies. Equiflow quantifies distributional shifts at each exclusion step and generates visualizations showing how key clinical and demographic variables evolve during participant selection. In a case study of sepsis patients from the eICU database, sequential exclusions reduced the sample from 126,750–1,094 patients. Requiring non-missing troponin measurements in the final step of data processing caused substantial demographic shifts that would typically remain invisible in traditional reporting. By making compositional biases visible during cohort construction before modeling begins, Equiflow enables researchers to make informed decisions about analyses and acknowledge limitations in generalizability to their readers. This standardized, open-source approach promotes transparency in clinical research and supports development of more equitable clinical AI systems, addressing a critical need as healthcare increasingly relies on data-driven decision making.

## Author summary

Medical research studies filter participants through multiple steps, often removing those with missing data, applying clinical criteria, or excluding based on demographic factors. While each step may seem routine, the cumulative

**Data availability statement:** The Equiflow source code is openly available at: https://github.com/MoreiraP12/equiflow-v2. The package can be found at the Python Package index: https://pypi.org/project/equiflow/. Access to the eICU Collaborative Research Database used in the case study is available to qualified researchers through PhysioNet (https://physionet.org/content/eicu-crd/) after completion of the required data use agreement.

**Funding:** The author(s) received no specific funding for this work.

**Competing interests:** The authors have declared that no competing interests exist.

effect can dramatically reshape who remains in the final dataset, introducing hidden biases that undermine study validity and generalizability. This problem is particularly concerning in AI applications, where algorithms learn directly from training data and can perpetuate healthcare disparities. We developed Equiflow, a free, open-source Python tool that automatically generates visual diagrams tracking how a study population changes at each filtering step. Unlike traditional reporting methods that show only participant counts, Equiflow reveals compositional shifts, such as whether excluding patients with missing lab values disproportionately removes certain demographic groups. We describe two case studies using real-world ICU data showing how routine exclusion criteria can alter fundamental characteristics of a cohort. These shifts, invisible in standard reporting, could affect which patients benefit from resulting clinical tools. By making such biases visible early in the research process, Equiflow enables researchers to make informed decisions and transparently acknowledge limitations in their findings.

## Introduction

Medical research often involves multiple steps of data processing, including screening participants for eligibility, removing entries with missing or corrupted data, and applying various inclusion and exclusion criteria. Although these steps are standard practice, each transformation can subtly or substantially alter a dataset's composition, fundamentally affecting its key characteristics and representativeness. While these changes to a dataset can be subtle, they may critically impact research findings.

Understanding these changes is especially critical in AI studies, where algorithms learn patterns and predictions directly from training data. If a dataset is not representative, for example due to removing entries at various stages of preprocessing, it can introduce structural biases into the resulting models. The selection problem can extend beyond AI studies as well. Even as efforts grow to make clinical study recruitment more inclusive and representative [1–4], a lack of attention to how data manipulations shape a final sample can undermine these efforts if selective exclusions continue to remain hidden.

Frequently, a final sample is described through a baseline table of demographic and clinical features (often the "Table 1") [5], but this fails to demonstrate how those characteristics relate to the original dataset or population from which the final sample was drawn. Conversely, flow diagrams like CONSORT [6] (for randomized controlled trials) or STROBE [7] (for observational studies) show changes in sample size between screening for eligibility and final cohort composition, but fail to capture how the composition of a sample changes as it goes through multiple steps of participant selection. Guidelines such as TRIPOD (for prediction model development and validation) [8], PROBAST+AI (for assessing quality, risk of bias, and applicability of prediction models) [9], and RECORD

**Table 1. Auxiliary table showing cohort characteristics across case study exclusion steps.**

| Variable | Category | Step 1 (n = 126,750) | Step 2 (n = 117,489) | Step 3 (n = 17,161) | Step 4 (n = 1,094) | SMD (1→2) | SMD (2→3) | SMD (3→4) |
|---|---|---|---|---|---|---|---|---|
| Age* | | 62.8 (16.2) | 62.5 (16.3) | 64.6 (15.5) | 69.1 (13.2) | 0 | 0.1 | 0.3 |
| APACHE IV score* | | 57.3 (25.8) | 56.2 (24.6) | 69.0 (26.2) | 76.3 (25.6) | 0 | 0.5 | 0.3 |
| Race / Ethnicity† | Caucasian | 98,333 (77.6) | 91,642 (78) | 13,387 (78.0) | 933 (85.3) | 0 | 0 | 0.2 |
| | African American | 14,705 (11.6) | 13,216 (11.3) | 1,810 (10.5) | 102 (9.3) | | | |
| | Other or Unknown | 5,979 (4.7) | 5,497 (4.7) | 822 (4.8) | 33 (3.0) | | | |
| | Hispanic | 4,714 (3.7) | 4,327 (3.7) | 666 (3.9) | 11 (1.0) | | | |
| | Asian | 2,152 (1.7) | 1,988 (1.7) | 333 (1.9) | 10 (0.9) | | | |
| | Native American | 867 (0.7) | 819 (0.7) | 143 (0.8) | 5 (0.5) | | | |
| Sex† | Female | 57,373 (45.3) | 55,564 (45.4) | 8,357 (48.7) | 511 (46.7) | 0 | 0.1 | 0 |
| | Male | 69,376 (54.7) | 66,655 (54.6) | 8,804 (51.3) | 583 (53.3) | | | |

* = mean (SD); † = n (%).

(for reporting studies using routinely collected health data) [10] emphasize the importance of transparent reporting of participant flow and data handling. PROBAST+AI in particular explicitly incorporates fairness and data representativeness into its signaling questions, yet none of these frameworks provide tools for visualizing compositional changes at each exclusion step.

To address these challenges, we build on our previously proposed participant flow diagram for AI studies [11] by introducing *Equiflow*, an open-source Python package that automates the creation of these standardized flow diagrams. Equiflow tracks cohort evolution at each major step in data processing by visually demonstrating changes in relevant sample-level sociodemographic and clinical factors. By visualizing the impact of inclusion/exclusion criteria and data cleaning, it also shines a light on hidden sources of selection bias that typical reporting formats overlook. This standardized approach aims to promote transparency and the development of fairer, more representative clinical AI models, and has been recommended as a method for depicting results in STROBE-Equity guidelines [12]. Although motivated by the needs of machine learning research, Equiflow can be useful to any study that filters participants using any inclusion and/or exclusion criteria, offering a transparent view of how a dataset is reshaped by these choices. In this paper, we present Equiflow's design and implementation, demonstrate its use through case studies, and discuss its potential to improve transparency in clinical research.

## Problems with exclusion criteria at various steps

While additional detail can be found in our original paper proposing this concept [11], we include a few brief examples below to illustrate the unintended consequences of not tracking participant exclusion steps throughout a study.

### Exclusion based on missingness

In many studies, patients with missing data are frequently excluded based on a predefined threshold [13]. Such missingness is often non-random: individuals with lower socioeconomic status or greater psychosocial needs are more likely to seek care from multiple facilities, leading to more sparse records in any single-center dataset [14,15]. Similarly, individuals from demographic groups who face structural barriers to care tend to receive less care overall, including fewer diagnostic tests, increasing the likelihood of missing information in the electronic health record [16]. Despite this potentially significant and systematic impact, many studies do not address missing data at all [17], let alone consider how that missingness may affect a model's ultimate output.

## Exclusion based on clinical characteristics

Defining a sample based on clinical characteristics is a double-edged sword; utilizing more restrictive criteria can increase internal validity at the expense of external validity, and vice versa. For example, when researchers applied inclusion and exclusion criteria from 158 antidepressant efficacy trials to 1,271 real-world outpatients with major depressive disorder, they found that a mean of 86.1% of patients would have been excluded, with exclusion rates worsening significantly over time (91.4% excluded in 2010–2014 versus 83.8% in 1995–2009) [18]. These findings indicate that the vast majority of depressed patients seen in clinical practice would not qualify for the trials meant to guide their treatment.

Additionally, utilizing clinical participation criteria that correlate with race, gender, or ethnicity can unintentionally introduce sampling bias. In a systematic review of pancreatic ductal adenocarcinoma clinical trials, traditional eligibility criteria based on chronic disease and multiple comorbidities disproportionately excluded Black patients compared to White patients [19]. These exclusions are problematic because they limit the generalizability and validity of trial findings to diverse populations and may not be medically necessary for ensuring trial safety.

## Exclusion based on demographic characteristics

Beyond clinical characteristics, demographic-based eligibility criteria can fundamentally alter a study's patient population. For example, language requirements, which are often established by researchers to streamline communication with participants, present a particularly significant barrier. A recent analysis of 309 thyroid cancer clinical trials found that 7.4% of trials excluded patients based on language; among trials that reported demographics, Asian, Black, and Hispanic patients were all significantly under-represented while White Americans were over-represented [20].

Similarly, upper age limits, often imposed due to concerns about comorbidities in older patients, can be especially problematic in clinical trials for conditions that predominantly affect older populations, such as spine-related disorders [21]. The cumulative effect of these demographic exclusions is that many studies fail to accurately represent their target patient populations, yielding findings that may not be observed in real-world clinical practice. Yet the effects of these exclusions on sample composition are rarely reported as part of the study results.

## Opportunities for intervention

Equiflow can help address the challenges above by providing visualizations of how each sample exclusion criterion reshapes cohort composition. This enables researchers to detect and describe these unintended shifts in the demographic or clinical characteristics of their sample and identify the risk of structural and statistical biases earlier in the research process. While it is not a direct solution to these biases, it offers visibility into patterns that might otherwise remain hidden and become embedded in downstream analyses.

To facilitate this discussion, we clarify two key terms used throughout this paper. We use "selection bias" to refer to the broader epidemiological concept of systematic error introduced when the study sample is not representative of the target population. We use "compositional bias" to refer specifically to shifts in the demographic or clinical makeup of a cohort resulting from sequential exclusion steps. Equiflow is designed to detect and visualize compositional bias, which is one mechanism through which selection bias can arise.

## Methods

### Overview and installation

Equiflow is an open-source Python package designed to create flow diagrams that track demographic and clinical changes throughout the participant selection process in research studies. The package requires Python 3.9 or later and can be installed from the Python Package Index using the command 'pip install equiflow.' The package accepts input data

as a Pandas DataFrame. Core dependencies of the package include Pandas, NumPy, Matplotlib, SciPy, and Graphviz, which are all automatically installed.

### User interface and workflow

We provide two interfaces to accommodate different user needs. The primary Equiflow class offers full control over all parameters and options as detailed below. For more rapid diagram creation, the EasyFlow wrapper class provides a simplified API with automatic variable type detection using data type inference.

The typical workflow consists of three phases: (1) initialization with variable specification, (2) sequential application of exclusion criteria in the order they occur in the study protocol, and (3) generation of outputs. As noted above, Equiflow requires explicit variable categorization, while EasyFlow can automatically detect variable types based on data characteristics. Both interfaces provide immediate visual feedback through the flow diagram.

### Core functionality

**Cohort tracking.** Researchers initialize the package with their raw dataset and specify variables of interest, which they can label as categorical, normally distributed continuous, or non-normally distributed continuous. The package then allows sequential application of exclusion criteria while maintaining snapshots of the cohort at each step.

Exclusions are applied using the "add_exclusion" method, which accepts either a boolean 'keep' parameter indicating which participants to retain, or a filtered dataframe containing the remaining cohort after that exclusion step is applied. Each exclusion requires a descriptive reason (and optionally a label) for the resulting cohort. The package validates that each exclusion reduces the cohort size and maintains data consistency across steps. Alternatively, the user can pass multiple pre-filtered dataframes along with exclusion labels into Equiflow, which then computes distributional changes between the consecutive dataframes.

**Statistical analysis.** For each cohort state, Equiflow automatically calculates descriptive statistics appropriate to each variable type. Categorical variables are summarized as counts and percentages, normally distributed continuous variables as mean ± standard deviation, and non-normally distributed continuous variables as median with interquartile range. To quantify compositional changes between cohorts, we implement standardized mean differences (SMDs), which represent differences in terms of standard deviations.

For continuous variables, the SMD is calculated as:

$$SMD = (\bar{x}_1 - \bar{x}_2 x) / \sqrt{[(s_1^2 + s_2^2)/2]}$$

where $\bar{x}_1$ and $\bar{x}_2$ are the means and $s_1^2$ and $s_2^2$ are the variances of the two cohorts being compared. For binary categorical variables, the SMD is computed using the difference in proportions:

$$SMD = (p_1 - p_2) / \sqrt{[p_1 (1 - p_1) + p_2(1 - p_2)/2]}$$

For multi-level categorical variables, we use the approach described by Yang and Dalton [22], which extends the SMD to multicategorical variables by computing the Mahalanobis distance between the proportion vectors of the two groups. To reduce small-sample bias, Equiflow also applies a Hedges' correction factor to all SMD calculations.

Based on the literature on inverse probability of treatment weighting [23], we suggest that an SMD greater than an absolute value of 0.1 standard deviations between steps may represent a meaningful change in sample composition. SMD has also been commonly bucketed into categories of |0.2| as a small effect size, |0.5| as a medium effect size, and |0.8| as a large effect size [24]. Notably, while these conventional effect size benchmarks of |0.2| (small), |0.5| (medium), and |0.8| (large) derived from Cohen's work [24] are frequently cited, these thresholds were originally developed for

experimental effect sizes in behavioral sciences and may not be universally appropriate for evaluating selection drift in observational cohorts.

Researchers should interpret SMD values with domain-specific context; a given effect size may be of greater or lesser consequence depending on the research question.

**Flow diagram and output generation.** The package generates flow diagrams using Graphviz for layout and Matplotlib for embedded distribution visualizations. Each exclusion step is represented as a directed graph showing sample sizes, exclusion nodes indicating removal reasons and counts, and distribution plots displaying key variable compositions.

Equiflow produces three primary tabular outputs: a flow summary table (participant counts at each step), a characteristics table (variable distributions per cohort), and a drift table (compositional changes expressed in SMDs). These elements are integrated visually in the flow diagram with embedded distribution plots. All tables are returned as Pandas Data-Frames, supporting standard export formats including CSV and LaTeX.

## Implementation details

**Variable specification and display.** Researchers can customize variable display through several parameters that enhance output interpretability. The "order_vars" parameter controls the sequence in which variables appear in all outputs, while "order_classes" specifies how categorical variable levels should be ordered within each variable. For categorical variables with many levels, the "limit" parameter restricts the number of displayed categories to avoid overcrowded visualizations. When applied, users specify the number (N) of categories to display; categories are ranked by frequency in the initial cohort, with the N most prevalent shown and all remaining categories aggregated into an "Other" group. Together, these features ensure that outputs remain interpretable even when working with complex datasets containing numerous variables and categories.

**Missing data handling, assumptions and limitations.** The Equiflow package implements unified missing data detection, standardizing various representations (empty strings, "NA", "None," etc.) to a consistent internal format. Missingness is tracked as a separate category for each variable, making patterns of missing data visible within flow visualizations. This design is intentional: rather than implementing missing data deletion or a default imputation strategy, Equiflow preserves missingness explicitly, allowing researchers to apply context-appropriate imputation if desired.

Notably, the interpretability of missingness as a category depends on the underlying missing data mechanism. Under conditions of Missing Completely at Random (MCAR), where missingness is unrelated to observed or unobserved data, visualizing missingness as a category or excluding missing cases is defensible [25]. Under Missing at Random (MAR), where missingness depends on observed covariates, the visualization remains descriptively informative but may not reflect the true distribution among complete cases. Under Missing Not at Random (MNAR), where missingness depends on unobserved values, treating missingness as a separate category can be misleading, as the missing group may be systematically biased [25].

Researchers using Equiflow should consider the likely missing data mechanism in their specific context. When substantial missingness exists and MAR or MNAR is plausible, we recommend: (1) conducting sensitivity analyses using multiple imputation to assess robustness of observed compositional shifts; (2) explicitly reporting the assumed missing data mechanism and its justification; and (3) acknowledging limitations in generalizability if missingness is likely informative. Equiflow does not implement built-in imputation, but researchers should apply these techniques upstream using established packages if needed.

**Software design and development.** The code is now openly available under the MIT License on GitHub (https://github.com/MoreiraP12/equiflow-v2), with the package available at the Python Package Index (PyPI): https://pypi.org/project/equiflow/. Notably, the Equiflow package is currently at version 0.1.10 and both case studies below were generated using this version of Equiflow.

**Software availability and maintenance.** Equiflow is distributed under the MIT License and is available through both GitHub and the PyPI. Breaking changes to the API will be documented in release notes and accompanied by migration

guides where applicable. The GitHub repository includes continuous integration testing via GitHub Actions, with a test suite comprising 51 tests (41 unit tests and 10 integration tests) executed automatically across Python versions 3.9 through 3.12 on each pull request. Documentation is hosted alongside the repository and includes API reference documentation, usage tutorials, and worked examples. Bug reports and feature requests can be submitted through GitHub Issues, with typical initial triage response times of 1 week for bug reports and 2 weeks for feature requests, reflecting the prioritization of fixing existing issues. Community contributions are welcomed through pull requests following the contribution guidelines documented in CONTRIBUTING.md.

**Quality assurance and testing.** All major functions were tested with example datasets representing common use cases in clinical research. Input validation ensures appropriate error messages for common issues such as incompatible exclusion criteria or missing variable specifications.

## Results

### First case study

To demonstrate the Equiflow package in action, we created two practical case studies. In the first case study, a hypothetical research team aims to investigate whether heart strain as assessed by elevated troponin levels (a cardiac biomarker indicating myocardial injury) is associated with increased mortality risk among sepsis patients in the intensive care unit (ICU) who were not admitted to the hospital for a presumed cardiac issue. They plan to use a logistic regression model to investigate this relationship. The team uses the eICU Collaborative Research Database (eICU CRD) as their study sample. The following sections detail their analytical approach and key findings.

### Problem definition

The question of whether myocardial injury (with elevated troponin as a proxy measurement) is an independent predictor of mortality for sepsis patients in the ICU (in patients not being admitted for a cardiac issue) is unanswered, with mixed evidence currently. While earlier meta-analyses reported significant mortality associations [26], a more recent 2025 analysis found elevated troponin had limited prognostic utility after adjustment for disease severity and confounders including any history of cardiac disease (adjusted OR 1.06, 95% CI: 0.99-1.13) [27].

Multiple studies have documented differences in cardiovascular testing and care across different demographic groups [28,29]. In fact, the largest analysis to date examining over 37 million emergency department visits demonstrates that men were more likely to receive troponin testing than women (OR 1.087), while Hispanic or Latino patients (OR 0.923) and non-white patients overall (OR range 0.918–0.950) were significantly less likely to be tested compared to white patients [30]. By contrast, patients who primarily spoke Spanish (OR 1.016) or other non-English languages (OR 1.064) were slightly more likely to undergo testing compared with English speakers [30].

A key methodological consideration in seeking to answer this question involves selection bias introduced by conditioning on troponin measurement availability. Troponin measurement availability here functions as a selection mechanism that can induce collider bias. Specifically, the decision to order troponin testing may be influenced by both patient characteristics (e.g., demographics, clinical presentation) and factors related to the outcome (e.g., clinical suspicion of cardiac involvement). When the analysis is restricted to solely patients with available troponin measurements, associations between patient characteristics and mortality can be distorted through the patient selection process. A directed acyclic graph (DAG) is shown in Fig 1 to visualize the assumed causal structure among the study variables.

### Dataset selection and processing workflow

As mentioned above, the team uses the eICU CRD which contains data from critical care units at over 200 hospitals across the United States [31]. The team decides to focus on a few key exclusion stages:

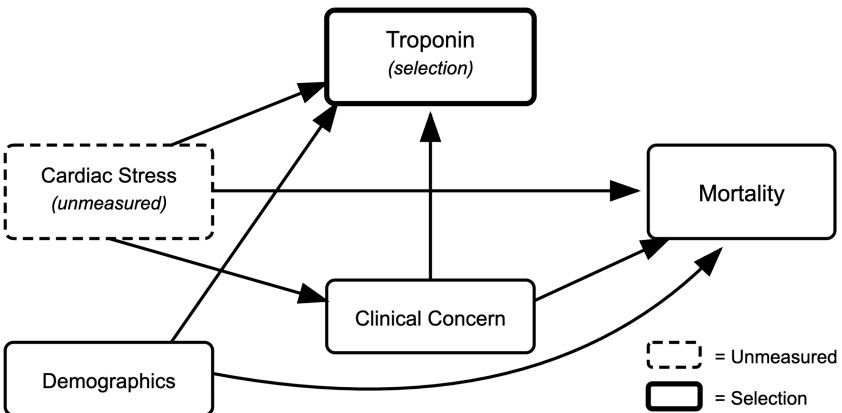

**Fig 1. Causal DAG depicting the assumed causal relationships among key variables in the research question.**

1. **Initial Recruitment**: Selection of all ICU patients in the eICU database

2. **Clinical Inclusion/Exclusion Criteria:**

   a. Exclusion Step 1: Excluding patients with known heart disease to focus on patients without established cardiac conditions.

   b. Exclusion Step 2: Limiting to patients with sepsis, as defined by the Acute Physiology and Chronic Health Evaluation (APACHE IV) admission diagnosis classification available in eICU.

3. **Outcome Availability Exclusion**

   a. Exclusion Step 3: Limiting to patients with available troponin measurements, which would serve as the primary exposure variable.

**Variable selection**

Researchers may be particularly worried about certain compositional changes. Given the findings above that the frequency of testing for troponin can vary between demographic groups [28–30], and that excluding patients with missing troponin data may disproportionately affect group representation, the team decided to track the distribution of sex, age, and ethnicity during dataset preprocessing. In addition, the team tracked each patient's maximum APACHE IV score over the hospitalization, a validated severity measure incorporating physiology, age, comorbidities, and admission diagnosis that effectively predicts hospital mortality [32]. This serves as a valuable "validity check": one would expect average maximum APACHE IV scores to rise after restricting the sample to only include sepsis patients (a severe systemic illness with multi-organ dysfunction), but otherwise remain stable across preprocessing steps. Substantial shifts in demographics or APACHE IV scores might suggest that the logistic regression results no longer generalize to the intended population.

**Equiflow implementation and findings**

The team generated an Equiflow diagram for this hypothetical study to monitor and visualize changes in cohort composition throughout each processing stage, with differences between stages tracked using the SMD metric (Fig 2). The first step produced minimal change in tracked variables, excluding only 9,261 of 126,750 patients from the eICU database. In the second step, filtering for the 17,161 (14.6%) patients with sepsis revealed a small increase in age but a more

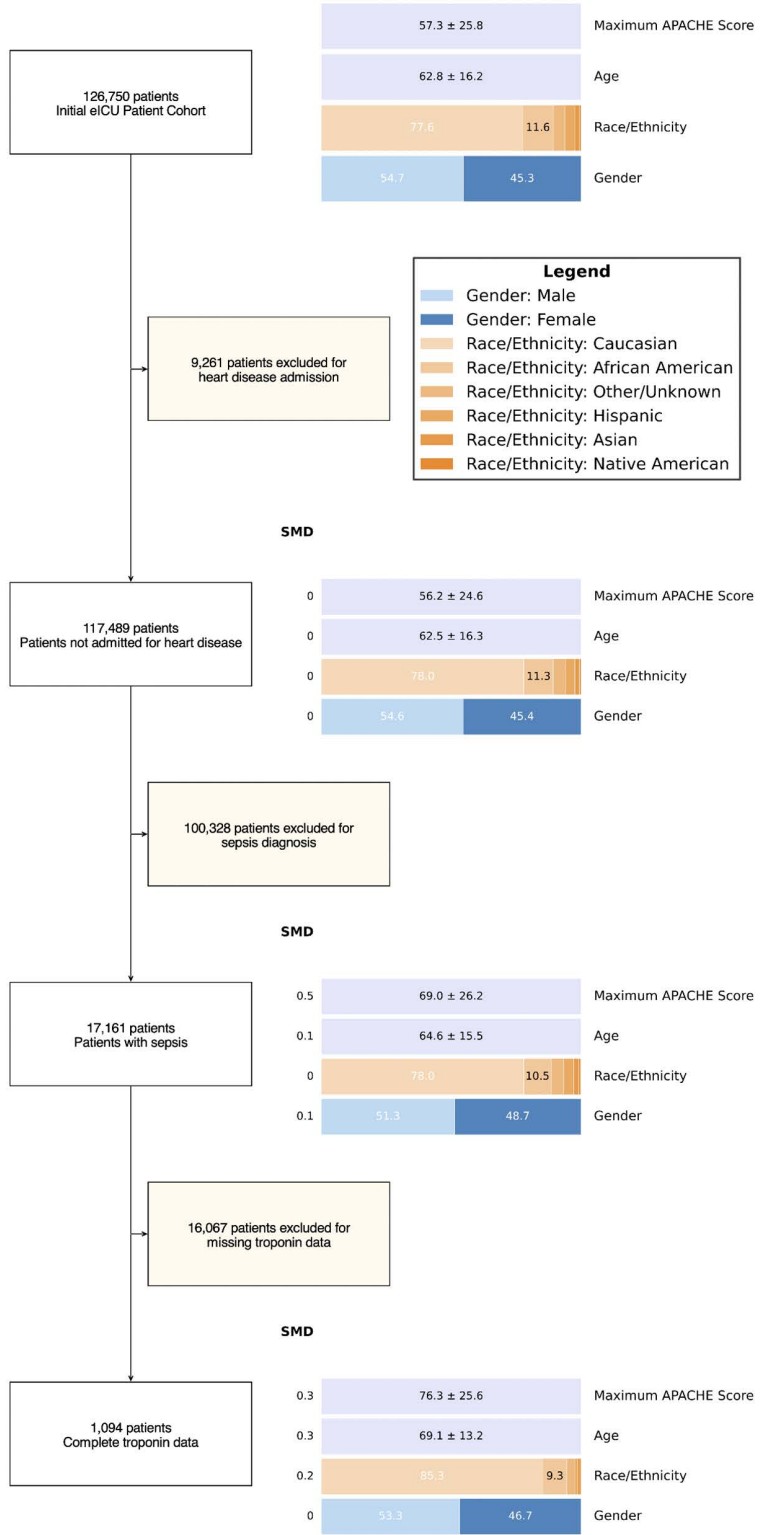

**Fig 2. Generated Equiflow diagram of selected case study.**

substantial increase in average maximum APACHE IV score from 56.2 to 69.0 (SMD = 0.5; a medium effect size), indicating a sicker population as expected when selecting for septic patients.

The final step, selecting only for the 1,094 (6.3%) patients who received troponin testing, produced the most pronounced shifts across all variables. Average age increased from 64.6 to 69.1 years (SMD = 0.3), and ethnicity distribution changed notably, with the proportion of white patients increasing from 78.0% to 85.3% (SMD = 0.2), while all other tracked ethnicities decreased. APACHE IV scores also increased again over this step to 76.3 (SMD = 0.3), which is consistent with selection induced by conditioning on troponin measurement availability, whereby patients with greater illness severity and suspicion of cardiac involvement are more likely to be tested. Notably, gender distribution remained largely stable throughout all processing steps.

We also show selected code used to generate the flow diagram (S1 Appendix), as well as a final table displaying these cohort sizes and shifts during the study (Table 1). This approach initialized EquiFlow with the full eICU dataset, specifying which variables to track as categorical (gender, ethnicity) versus continuous (age, max_apache_score). The library then applies exclusion criteria sequentially through the 'add_exclusion' method. Each exclusion preserves the data state, allowing Equiflow to compute distributional changes between consecutive stages. The final 'plot_flows' function generates the comprehensive visualization, rendering both the patient flow counts and the distribution plots for tracked variables. The function parameters control visual aspects including box dimensions, color schemes for categorical variables, and whether to display SMD values. The full code used to generate this diagram is available at: https://github.com/MoreiraP12/equiflow-v2/blob/master/tests/test_eicu_integration.py. For users without PhysioNet access, the GitHub repository includes a synthetic data generator that produces eICU-like data with realistic demographic distributions for testing purposes.

## Second case study

Given the extreme exclusion rate in the final step of the first case study (93.6% of sepsis patients excluded based on troponin availability), we present a second case study using the Medical Information Mart for Intensive Care IV (MIMIC-IV) v3.1 database [33] to demonstrate Equiflow's utility in a research scenario involving more modest exclusion steps. In this hypothetical study, a research team aims to investigate whether elevated creatinine on admission is associated with hospital mortality among ICU patients. Acute kidney injury (AKI), for which creatinine is a key biomarker, is a well-established risk factor for mortality in critically ill populations [34]. The team begins with all 94,458 ICU visits in MIMIC-IV and applies two exclusion steps. First, they restrict to each patient's first ICU stay in the dataset only (excluding 29,092 patients with multiple ICU admissions), yielding 65,366 patients. Second, they exclude 4,006 patients with missing creatinine values or ICU stays shorter than 12 hours (requiring adequate observation windows for first-day clinical data), producing a final analytic cohort of 61,360 patients.

The team generated an Equiflow diagram (Fig 3) and auxiliary table (Table 2) to track compositional changes across these steps. In contrast to the first case study, where the final troponin-based exclusion produced large demographic shifts, this scenario demonstrates smaller exclusion effects.

The first exclusion step, restricting the cohort to first ICU stays, produced an insurance SMD of 0.07. This shift was driven by a decrease in Medicare representation from 54.9% to 52.7% and an increase in private insurance from 26.0% to 28.2%, indicating that patients with Medicare were more likely to have multiple hospitalizations, whereas privately insured patients were less likely to do so. Sex and age distributions remained essentially unchanged. The second exclusion step, requiring complete creatinine data and an adequate length of stay, produced uniformly small SMDs across all tracked variables (all ≤ 0.03). ICU mortality declined modestly across steps, from 8.0% to 7.5% to 6.7%, consistent with the exclusion of shorter ICU stays that may represent more acute or rapidly fatal presentations.

This case study also demonstrates an alternative workflow for using Equiflow, in which users load pre-filtered DataFrames rather than applying exclusion criteria within the package itself. Here, the team exported datasets at each

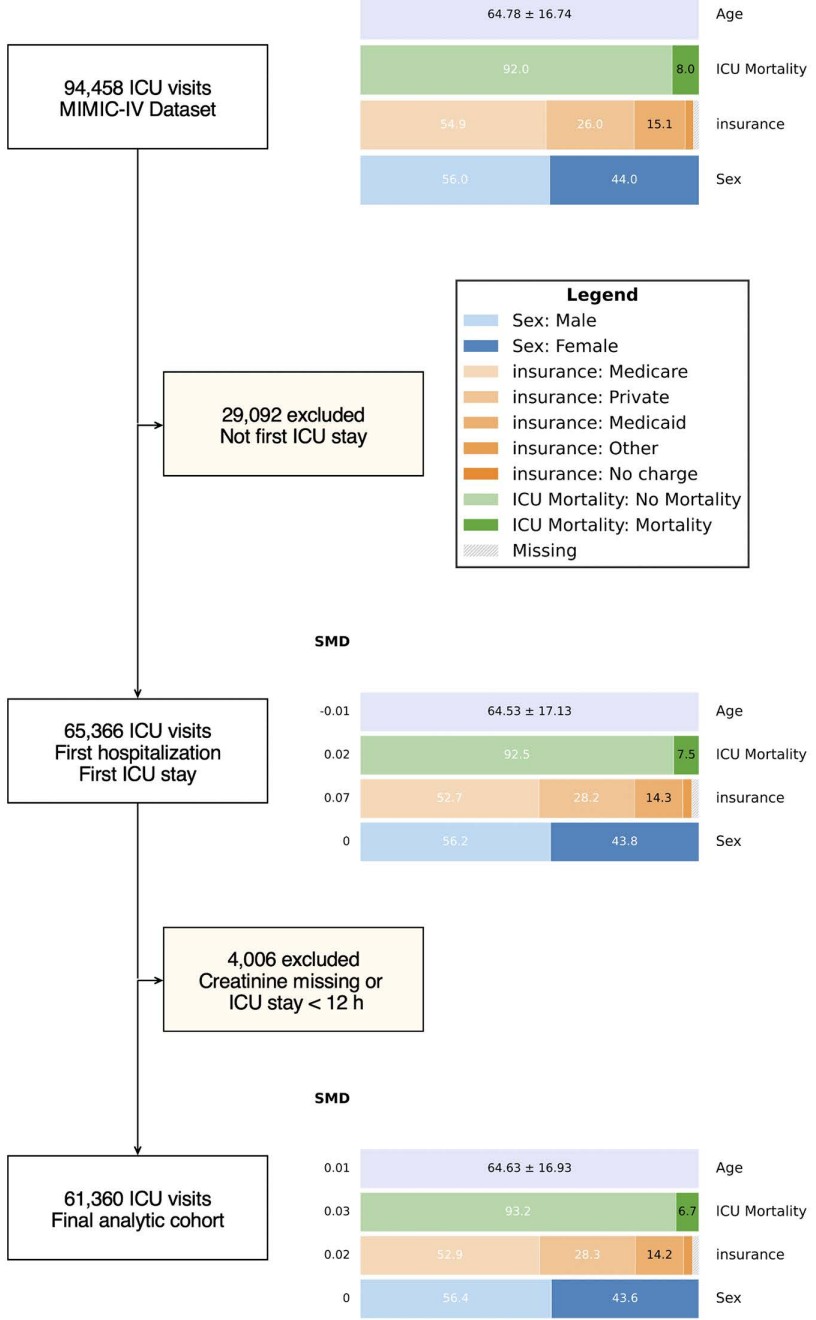

**Fig 3. Equiflow diagram for the MIMIC-IV case study showing cohort evolution across two exclusion steps.**

exclusion step and passed them directly to Equiflow as a list of DataFrames, with Equiflow computing distributional changes between consecutive stages. This approach can be a useful alternative when preprocessing is performed in a separate environment or when exclusion logic is complex. Selected code for this case study is also displayed in the S1 Appendix.

**Table 2. Cohort characteristics across MIMIC-IV case study exclusion steps.**

| Variable | Category | Step 1 (n=94,458) | Step 2 (n=65,366) | Step 3 (n=61,360) | SMD (1→2) | SMD (2→3) |
|---|---|---|---|---|---|---|
| Age* | | 64.78 (16.74) | 64.53 (17.13) | 64.63 (16.93) | -0.01 | 0.01 |
| Sex | Male | 52,875 (55.98) | 36,720 (56.18) | 34,608 (56.40) | | |
| Sex | Female | 41,583 (44.02) | 28,646 (43.82) | 26,739 (43.58) | 0.0 | 0.0 |
| Insurance | Medicare | 51,819 (54.86) | 34,464 (52.72) | 32,457 (52.90) | | |
| Insurance | Private | 24,540 (25.98) | 18,447 (28.22) | 17,383 (28.33) | | |
| Insurance | Medicaid | 14,240 (15.08) | 9,353 (14.31) | 8,733 (14.23) | | |
| Insurance | Other | 2,328 (2.46) | 1,733 (2.65) | 1,633 (2.66) | | |
| Insurance | No charge | 8 (0.01) | 7 (0.01) | 7 (0.01) | | |
| Insurance | Missing | 1,523 (1.61) | 1,362 (2.08) | 1,147 (1.87) | 0.07 | 0.02 |
| ICU Mortality | No Mortality | 86,930 (92.03) | 60,485 (92.53) | 57,209 (93.24) | | |
| ICU Mortality | Mortality | 7,528 (7.97) | 4,881 (7.47) | 4,138 (6.74) | 0.02 | 0.03 |

*Reported as Mean (SD). Categorical variables reported as N (%). SMD=standardized mean difference.

## Discussion

These case studies demonstrate Equiflow's ability to reveal hidden patterns of selection bias that would otherwise remain invisible in traditional reporting formats. In the eICU case study, the requirement for troponin measurements in the final exclusion step led to substantial demographic shifts, including large increases in age and APACHE IV score and an increase in white patient representation by 7.3 percentage points, that reflect documented disparities in cardiovascular testing [28,29]. These shifts also illustrate the consequences of conditioning on a variable that may function as a collider in a causal structure linking patient characteristics, clinical decision-making, and outcomes. In contrast, the MIMIC-IV case study demonstrates that even when individual exclusion steps produce small compositional shifts (all SMDs ≤ 0.07), Equiflow can provide a transparent record of any potential changes that would otherwise go unreported.

By automating the creation of these flow diagrams, the package may help alter what can often be an overlooked aspect of research into a more standardized, reproducible component of the scientific workflow. This standardization is highly valuable: while individual researchers might occasionally examine demographic shifts manually, Equiflow has the potential to make such examination more routine and systematic across studies that adopt it [35].

Furthermore, we believe that Equiflow's real value lies in its timing, as it identifies compositional biases at the point of cohort construction, before any modeling or analysis begins. This early detection enables researchers to make informed decisions about whether to proceed with their planned analysis, implement corrective measures, or simply acknowledge specific limitations in the generalizability of their findings. Without such tools, biased datasets can propagate through the entire research pipeline, ultimately producing clinical decision support systems that perpetuate or amplify existing healthcare disparities [36,37].

Several important limitations of Equiflow warrant consideration. First, when exclusions involve multiple characteristics simultaneously or equity factors with more than two levels (e.g., intersecting factors like poverty and rurality), the visual flow diagram can become more challenging to interpret. Second, the package is currently only available in Python, potentially limiting adoption among researchers who primarily use R or other statistical software. The development of an R package is a priority for future development. Third, while Equiflow makes biases visible, it does not automatically suggest corrective actions or provide built-in bias mitigation methods. Researchers must determine how to address identified biases, whether through sampling strategies, statistical weighting, or acknowledgment of limitations for generalizability. Fourth, the package's default treatment of missingness as a category is most appropriate under MCAR assumptions and may require supplementation with sensitivity analyses under MAR or MNAR mechanisms.

As clinical research increasingly relies on machine learning and AI, we hope that tools like Equiflow can add to the infrastructure for responsible research. The package not only seeks to improve transparency in reporting but actively contributes to the development of more accurate healthcare technologies by making invisible biases visible and highlighting potential limits to the generalizability of findings. Its open-source nature and ease of integration into existing Python workflows lower barriers to adoption, potentially establishing a new standard for how cohort selection is documented and evaluated in clinical research.

## Supporting information

**S1 Appendix. Code examples demonstrating cohort initialization, application of exclusion criteria, and generation of flow diagrams using EquiFlow and EasyFlow.**
(DOCX)

## Author contributions

**Conceptualization:** Jacob Gould Ellen, Martin Viola, Arinda Jordan, João Matos, Leo Anthony Celi.

**Data curation:** Jacob Gould Ellen, Chrystinne Oliveira Fernandes, Martin Viola, Arinda Jordan, João Matos, Pedro Moreira.

**Formal analysis:** Jacob Gould Ellen, Chrystinne Oliveira Fernandes, Martin Viola, Pedro Moreira.

**Investigation:** Jacob Gould Ellen, Martin Viola, Keagan Yap, Mutesi Flavia Kirabo, Leo Anthony Celi.

**Methodology:** Jacob Gould Ellen, Martin Viola, Arinda Jordan, Mutesi Flavia Kirabo.

**Project administration:** Arinda Jordan, Leo Anthony Celi.

**Resources:** João Matos.

**Software:** Jacob Gould Ellen, Chrystinne Oliveira Fernandes, João Matos, Pedro Moreira.

**Supervision:** Martin Viola, Leo Anthony Celi.

**Validation:** Chrystinne Oliveira Fernandes, Pedro Moreira.

**Visualization:** Jacob Gould Ellen, Chrystinne Oliveira Fernandes, João Matos.

**Writing – original draft:** Jacob Gould Ellen, Martin Viola, Keagan Yap.

**Writing – review & editing:** Jacob Gould Ellen, Chrystinne Oliveira Fernandes, Martin Viola, João Matos, Leo Anthony Celi.

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
