## [Decision Letter · Decision Letter 0]

3 Nov 2025

Response to Reviewers'. This file does not need to include responses to any formatting updates and technical items listed in the 'Journal Requirements' section below.'. This file does not need to include responses to any formatting updates and technical items listed in the 'Journal Requirements' section below.* A marked-up copy of your manuscript that highlights changes made to the original version. You should upload this as a separate file labeled 'Revised Manuscript with Track Changes'.'.* An unmarked version of your revised paper without tracked changes. You should upload this as a separate file labeled 'Manuscript'.'. If you would like to make changes to your financial disclosure, competing interests statement, or data availability statement, please make these updates within the submission form at the time of resubmission. Guidelines for resubmitting your figure files are available below the reviewer comments at the end of this letter. We look forward to receiving your revised manuscript. Kind regards, Po-Chih Kuo, Ph. D.Section EditorPLOS Digital Health Po-Chih KuoSection EditorPLOS Digital Health Leo Anthony CeliEditor-in-ChiefPLOS Digital Healthorcid.org/0000-0001-6712-6626 **Journal Requirements:** 

1. Please upload separate figure files in .tif or .eps format. Also, remove the figures from your manuscript file but keep the legends.

2. Please provide an Author Summary. This should appear in your manuscript between the Abstract (if applicable) and the Introduction, and should be 150–200 words long. The aim should be to make your findings accessible to a wide audience that includes both scientists and non-scientists. Sample summaries can be found on our website under Submission Guidelines:

https://journals.plos.org/digitalhealth/s/submission-guidelines#loc-parts-of-a-submission

**Additional Editor Comments (if provided):****Reviewers' Comments:** Reviewer's Responses to Questions

**Comments to the Author**

1. Does this manuscript meet PLOS Digital Health’s publication criteria? Is the manuscript technically sound, and do the data support the conclusions? The manuscript must describe methodologically and ethically rigorous research with conclusions that are appropriately drawn based on the data presented.? Is the manuscript technically sound, and do the data support the conclusions? The manuscript must describe methodologically and ethically rigorous research with conclusions that are appropriately drawn based on the data presented.

Reviewer #1: Yes

Reviewer #2: Partly

2. Has the statistical analysis been performed appropriately and rigorously?

Reviewer #1: Yes

Reviewer #2: No

3. Have the authors made all data underlying the findings in their manuscript fully available (please refer to the Data Availability Statement at the start of the manuscript PDF file)?

The PLOS Data policy requires authors to make all data underlying the findings described in their manuscript fully available without restriction, with rare exception. The data should be provided as part of the manuscript or its supporting information, or deposited to a public repository. For example, in addition to summary statistics, the data points behind means, medians and variance measures should be available. If there are restrictions on publicly sharing data—e.g. participant privacy or use of data from a third party—those must be specified.requires authors to make all data underlying the findings described in their manuscript fully available without restriction, with rare exception. The data should be provided as part of the manuscript or its supporting information, or deposited to a public repository. For example, in addition to summary statistics, the data points behind means, medians and variance measures should be available. If there are restrictions on publicly sharing data—e.g. participant privacy or use of data from a third party—those must be specified.

Reviewer #1: Yes

Reviewer #2: Yes

4. Is the manuscript presented in an intelligible fashion and written in standard English?

Reviewer #1: Yes

Reviewer #2: No

Reviewer #1: This manuscript introduces Equiflow, an open-source Python package that automates the creation of enhanced participant flow diagrams to track both sample size and demographic/clinical composition changes throughout data preprocessing in clinical research studies. The authors demonstrate the tool's utility through a sepsis cohort case study using the eICU database, revealing how standard exclusion criteria can introduce substantial demographic shifts that would remain invisible in traditional reporting formats.

1. Originality

The work represents a significant advancement beyond the authors' previous conceptual framework (Ellen et al., 2024, J Biomed Inform). While that paper proposed the theoretical approach, this manuscript delivers a practical, implemented solution that researchers can immediately use. The automated tracking of compositional changes using standardized mean differences (SMD), integrated with visual flow diagrams, represents a novel contribution to research transparency tools. The combination of traditional flow diagram elements with distributional visualizations and quantitative drift metrics has not been previously implemented in an accessible, open-source format.

2. High Importance and Broad Interest

This tool addresses a critical gap in digital health research methodology. As clinical AI/ML systems increasingly influence healthcare decisions, understanding and documenting selection biases during cohort construction becomes essential for developing fair and generalizable models. The case study's finding that requiring troponin measurements increased white patient representation by 7.3 percentage points exemplifies how routine data requirements can introduce substantial demographic biases - a finding with immediate implications for the validity of cardiovascular risk models.

3. Methodological Rigor and Ethical Standards

The methodology is sound:

Appropriate use of SMD for quantifying distributional shifts

Clear variable categorization (categorical, normal continuous, non-normal continuous)

Proper handling of missing data

Well-documented software engineering practices (version control, testing, documentation)

Appropriate use of publicly available eICU data with proper citations

MIT licensing ensuring broad accessibility

The statistical approach using SMD with a threshold of |0.1| is well-justified from the propensity score literature, though the authors appropriately acknowledge this may be context-dependent.

4. Evidence Supporting Conclusions

The case study effectively demonstrates the tool's capability. The progression from 126,750 patients to 1,094, with substantial demographic shifts at the troponin testing requirement step, convincingly illustrates how standard preprocessing can alter cohort composition. The evidence directly supports the authors' claim that Equiflow can reveal hidden patterns of exclusion.

5. Utility and Accessibility

Exceptional accessibility.

Reviewer #2: Equiflow tackles a valuable problem and the tool has strong practical potential, but the manuscript as written omits critical statistical detail and makes misleading causal claims; these must be corrected to meet publication standards. After the authors supply the requested mathematical definitions, sensitivity analyses, clearer causal framing, and improved documentation, the paper could make a significant contribution to transparent cohort reporting.

**Do you want your identity to be public for this peer review?** For information about this choice, including consent withdrawal, please see our Privacy Policy..

Reviewer #1: No

Reviewer #2: **Yes:** Perkins WatambwaPerkins WatambwaPerkins WatambwaPerkins Watambwa

**Figure resubmission:** While revising your submission, we strongly recommend that you use PLOS’s NAAS tool (https://ngplosjournals.pagemajik.ai/artanalysis) to test your figure files. NAAS can convert your figure files to the TIFF file type and meet basic requirements (such as print size, resolution), or provide you with a report on issues that do not meet our requirements and that NAAS cannot fix.

**Reproducibility:** To enhance the reproducibility of your results, we recommend that authors of applicable studies deposit laboratory protocols in protocols.io, where a protocol can be assigned its own identifier (DOI) such that it can be cited independently in the future. Additionally, PLOS ONE offers an option to publish peer-reviewed clinical study protocols. Read more information on sharing protocols at https://plos.org/protocols?utm_medium=editorial-email&utm_source=authorletters&utm_campaign=protocols To enhance the reproducibility of your results, we recommend that authors of applicable studies deposit laboratory protocols in protocols.io, where a protocol can be assigned its own identifier (DOI) such that it can be cited independently in the future. Additionally, PLOS ONE offers an option to publish peer-reviewed clinical study protocols. Read more information on sharing protocols at https://plos.org/protocols?utm_medium=editorial-email&utm_source=authorletters&utm_campaign=protocols

---

## [Decision Letter · Decision Letter 1]

26 Feb 2026

Response to Reviewers'. This file does not need to include responses to any formatting updates and technical items listed in the 'Journal Requirements' section below.'. This file does not need to include responses to any formatting updates and technical items listed in the 'Journal Requirements' section below.* A marked-up copy of your manuscript that highlights changes made to the original version. You should upload this as a separate file labeled 'Revised Manuscript with Track Changes'.'.* An unmarked version of your revised paper without tracked changes. You should upload this as a separate file labeled 'Manuscript'.'. If you would like to make changes to your financial disclosure, competing interests statement, or data availability statement, please make these updates within the submission form at the time of resubmission. Guidelines for resubmitting your figure files are available below the reviewer comments at the end of this letter. We look forward to receiving your revised manuscript. Kind regards, Po-Chih Kuo, Ph. D.Section EditorPLOS Digital Health Po-Chih KuoSection EditorPLOS Digital Health Leo Anthony CeliEditor-in-ChiefPLOS Digital Healthorcid.org/0000-0001-6712-6626  **Journal Requirements:** If the reviewer comments include a recommendation to cite specific previously published works, please review and evaluate these publications to determine whether they are relevant and should be cited. There is no requirement to cite these works unless the editor has indicated otherwise.  **Additional Editor Comments (if provided):****Reviewers' Comments:** Reviewer's Responses to Questions

**Comments to the Author**

Reviewer #1: All comments have been addressed

Reviewer #2: All comments have been addressed

publication criteria? Is the manuscript technically sound, and do the data support the conclusions? The manuscript must describe methodologically and ethically rigorous research with conclusions that are appropriately drawn based on the data presented.? Is the manuscript technically sound, and do the data support the conclusions? The manuscript must describe methodologically and ethically rigorous research with conclusions that are appropriately drawn based on the data presented.

Reviewer #1: Yes

Reviewer #2: Yes

3. Has the statistical analysis been performed appropriately and rigorously?

Reviewer #1: Yes

Reviewer #2: Yes

4. Have the authors made all data underlying the findings in their manuscript fully available (please refer to the Data Availability Statement at the start of the manuscript PDF file)?

The PLOS Data policy requires authors to make all data underlying the findings described in their manuscript fully available without restriction, with rare exception. The data should be provided as part of the manuscript or its supporting information, or deposited to a public repository. For example, in addition to summary statistics, the data points behind means, medians and variance measures should be available. If there are restrictions on publicly sharing data—e.g. participant privacy or use of data from a third party—those must be specified.requires authors to make all data underlying the findings described in their manuscript fully available without restriction, with rare exception. The data should be provided as part of the manuscript or its supporting information, or deposited to a public repository. For example, in addition to summary statistics, the data points behind means, medians and variance measures should be available. If there are restrictions on publicly sharing data—e.g. participant privacy or use of data from a third party—those must be specified.

Reviewer #1: Yes

Reviewer #2: Yes

5. Is the manuscript presented in an intelligible fashion and written in standard English?

Reviewer #1: Yes

Reviewer #2: Yes

Reviewer #1: (No Response)

Reviewer #2: Equiflow Manuscript (PDIG-D-25-00774R1)

This manuscript presents Equiflow, an open-source Python package for generating enhanced participant flow diagrams that track compositional changes in cohort characteristics during data preprocessing. The tool addresses a genuine gap in research transparency by visualizing how exclusion criteria alter demographic and clinical distributions beyond simple sample size reductions. The revision has substantially addressed previous concerns regarding SMD definitions, missing data assumptions, causal framing, and statistical testing.

Major Concerns Requiring Attention

1. Statistical Testing Section Contradictions

The manuscript states that statistical tests serve a "descriptive rather than an inferential purpose" but then describes automated hypothesis testing with p-value thresholds (p < 0.05 for normality rejection). This creates conceptual confusion:

- If tests are descriptive, why implement formal hypothesis testing with alpha thresholds?

- The recommendation to "prioritize effect sizes over p-values" conflicts with the package's automated test selection based on normality p-values.

Clarify the philosophical stance. Either (a) remove formal hypothesis testing entirely and rely solely on SMDs, or (b) acknowledge that tests serve a screening function and provide clearer guidance on interpretation.

2. Case Study Generalizability

The case study uses eICU data where the final exclusion step (requiring troponin measurement) reduces the sample from 17,161 to 1,094 (93.6% exclusion). This extreme reduction may not be representative of typical exclusion scenarios. The manuscript would benefit from:

- A second, less extreme case study demonstrating utility in more common scenarios

- Discussion of how Equiflow performs when exclusions are more modest (e.g., 10-20% reduction)

Minor Concerns

1. Software Engineering Details

- The manuscript mentions "continuous integration testing via GitHub Actions" but does not report test coverage metrics. For a methods paper, reporting coverage percentage would strengthen confidence in software reliability.

- The statement "typical response times of 1-2 weeks" for issue triage is vague. Consider specifying whether this applies to bug reports, feature requests, or both.

2. Reproducibility

- The case study code is available on GitHub, but the manuscript does not specify the exact package version used. Given that the package is under active development, version pinning is essential for reproducibility.

- The eICU data access requires credentialing; consider providing a synthetic dataset for users to test the package without PhysioNet access.

3. Terminology Consistency

- The manuscript alternates between "selection bias" and "compositional bias" without clear distinction. Consider defining these terms explicitly in the Methods section.

4. Figure Quality

- Figure 2 (code example) appears to be a just a typed code or screenshot/image of code rather than properly formatted code. For a software methods paper, code should be presented in a monospace font with syntax highlighting if possible, or moved entirely to supplementary materials with proper formatting.

5. Literature Gaps

- The manuscript does not cite the RECORD guidelines (Benchimol et al., 2015) for reporting studies using routinely collected health data, which are highly relevant.

- No mention of the PROBAST-AI extension currently under development, which may supersede some recommendations.

Minor Typographical/Formatting Issues

1. Reference 7 (STROBE) has an incorrect DOI pointing to a different publication (10.1016/j.ijsu.2014.07.014 is for International Journal of Surgery, not PLoS Medicine).

**Do you want your identity to be public for this peer review?** For information about this choice, including consent withdrawal, please see our Privacy Policy..

Reviewer #1: No

Reviewer #2: **Yes:** Perkins WatambwaPerkins WatambwaPerkins WatambwaPerkins Watambwa

**Figure resubmission:** While revising your submission, we strongly recommend that you use PLOS’s NAAS tool (https://ngplosjournals.pagemajik.ai/artanalysis) to test your figure files. NAAS can convert your figure files to the TIFF file type and meet basic requirements (such as print size, resolution), or provide you with a report on issues that do not meet our requirements and that NAAS cannot fix.

**Reproducibility:** To enhance the reproducibility of your results, we recommend that authors of applicable studies deposit laboratory protocols in protocols.io, where a protocol can be assigned its own identifier (DOI) such that it can be cited independently in the future. Additionally, PLOS ONE offers an option to publish peer-reviewed clinical study protocols. Read more information on sharing protocols at https://plos.org/protocols?utm_medium=editorial-email&utm_source=authorletters&utm_campaign=protocols To enhance the reproducibility of your results, we recommend that authors of applicable studies deposit laboratory protocols in protocols.io, where a protocol can be assigned its own identifier (DOI) such that it can be cited independently in the future. Additionally, PLOS ONE offers an option to publish peer-reviewed clinical study protocols. Read more information on sharing protocols at https://plos.org/protocols?utm_medium=editorial-email&utm_source=authorletters&utm_campaign=protocols

---

## [Decision Letter · Decision Letter 2]

18 Mar 2026

Equiflow: An open-source software package for evaluating changes in cohort composition

PDIG-D-25-00774R2

Dear Dr. Fernandes,

We are pleased to inform you that your manuscript 'Equiflow: An open-source software package for evaluating changes in cohort composition' has been provisionally accepted for publication in PLOS Digital Health.

Best regards,

Po-Chih Kuo, Ph. D.

Section Editor

PLOS Digital Health

**Additional Editor Comments (if provided):**

**Reviewer Comments (if any, and for reference):**

Reviewer's Responses to Questions

**Comments to the Author**

Reviewer #2: All comments have been addressed

publication criteria? Is the manuscript technically sound, and do the data support the conclusions? The manuscript must describe methodologically and ethically rigorous research with conclusions that are appropriately drawn based on the data presented.? Is the manuscript technically sound, and do the data support the conclusions? The manuscript must describe methodologically and ethically rigorous research with conclusions that are appropriately drawn based on the data presented.

Reviewer #2: Yes

3. Has the statistical analysis been performed appropriately and rigorously?

Reviewer #2: Yes

4. Have the authors made all data underlying the findings in their manuscript fully available (please refer to the Data Availability Statement at the start of the manuscript PDF file)?

The PLOS Data policy requires authors to make all data underlying the findings described in their manuscript fully available without restriction, with rare exception. The data should be provided as part of the manuscript or its supporting information, or deposited to a public repository. For example, in addition to summary statistics, the data points behind means, medians and variance measures should be available. If there are restrictions on publicly sharing data—e.g. participant privacy or use of data from a third party—those must be specified.requires authors to make all data underlying the findings described in their manuscript fully available without restriction, with rare exception. The data should be provided as part of the manuscript or its supporting information, or deposited to a public repository. For example, in addition to summary statistics, the data points behind means, medians and variance measures should be available. If there are restrictions on publicly sharing data—e.g. participant privacy or use of data from a third party—those must be specified.

Reviewer #2: Yes

5. Is the manuscript presented in an intelligible fashion and written in standard English?

PLOS Digital Health does not copyedit accepted manuscripts, so the language in submitted articles must be clear, correct, and unambiguous. Any typographical or grammatical errors should be corrected at revision, so please note any specific errors here.

Reviewer #2: Yes

Reviewer #2: I find the work to be substantially improved and methodologically sound. The authors have been exceptionally responsive to prior reviewer feedback, addressing each concern with thoughtful revisions. This manuscript presents a well-conceived, technically sound, and practically useful open-source tool that addresses a genuine gap in clinical research transparency. The revisions have strengthened the work considerably.

**Do you want your identity to be public for this peer review?** For information about this choice, including consent withdrawal, please see our Privacy Policy..

Reviewer #2: **Yes:** Perkins WatambwaPerkins WatambwaPerkins WatambwaPerkins Watambwa
